# Factors that May Protect the Native Hibernator Syrian Hamster Renal Tubular Epithelial Cells from Ferroptosis Due to Warm Anoxia-Reoxygenation

**DOI:** 10.3390/biology8020022

**Published:** 2019-03-31

**Authors:** Theodoros Eleftheriadis, Georgios Pissas, Vassilios Liakopoulos, Ioannis Stefanidis

**Affiliations:** Department of Nephrology, Faculty of Medicine, University of Thessaly, 41110 Larissa, Greece; gpissas@msn.com (G.P.); liakopul@otenet.gr (V.L.); stefanid@med.uth.gr (I.S.)

**Keywords:** hibernation, ischemia-reperfusion, ferroptosis, xCT, ferritin, GPX4

## Abstract

Warm anoxia-reoxygenation induces ferroptotic cell death in mouse proximal renal tubular epithelial cells (RPTECs), whereas RPTECs of the native hibernator Syrian hamster resist cell death. Clarifying how hamster cells escape ferroptosis may reveal new molecular targets for preventing or ameliorating ischemia-reperfusion-induced human diseases or expanding the survival of organ transplants. Mouse or hamster RPTECs were subjected to anoxia and subsequent reoxygenation. Cell death was assessed with the lactated dehydrogenase (LDH) release assay and lipid peroxidation by measuring cellular malondialdehyde (MDA) fluorometrically. The effect of the ferroptosis inhibitor α-tocopherol on cell survival was assessed by the 2,3-bis (2-methoxy-4-nitro-5-sulfophenyl)-5-[(phenylamino) carbonyl]-2H-tetrazolium hydroxide (XTT) assay. The expression of the critical ferroptotic elements cystine-glutamate antiporter (xCT), ferritin, and glutathione peroxidase 4 (GPX4) was assessed by Western blot. Contrary to mouse RPTECs, hamster RPTECs resisted anoxia-reoxygenation-induced cell death and lipid peroxidation. In mouse RPTECs, α-tocopherol increased cell survival. Anoxia increased the levels of xCT, ferritin, and GPX4 in both cell types. During reoxygenation, at which reactive oxygen species overproduction occurs, xCT and ferritin decreased, whereas GPX4 increased in mouse RPTECs. In hamster RPTECs, reoxygenation raised xCT and ferritin, but lowered GPX4. Hamster RPTECs resist lipid peroxidation-induced cell death. From the three main evaluated components of the ferroptotic pathway, the increased expression of xCT and ferritin may contribute to the resistance of the hamster RPTECs to warm anoxia-reoxygenation.

## 1. Introduction

Ischemia-reperfusion (I-R) injury plays a significant role in the pathogenesis of many human diseases such as myocardial infarction or stroke due to an occluded artery, or multiorgan failure due to decreased effective blood volume [1,2,3]. The kidney is also vulnerable to I-R injury, with the latter being the primary cause of acute kidney injury (AKI) [4]. Furthermore, I-R is responsible for the deterioration of organ transplants [5]. Ischemic tissue injury results from oxygen and nutrient deprivation. However, tissue reperfusion due to revascularization of an occluded artery, restoration of effective blood volume, or transplantation of an organ graft, may exacerbate the tissue damage through the production reactive oxygen species (ROS) by the massive re-entry of oxygen and nutrients [1,2,5].

In the animal kingdom, there are paradigms of resistance to I-R injury and study of the molecular mechanisms that govern them may reveal clinically useful information. Many mammals fall into a state of hibernation during the winter months to survive this period of food shortage. Hibernation is characterized by long periods of deep torpor with a dramatic drop in body temperature, heart and breathing rate, and blood pressure that an inexperienced observer may think that the animal is dead. These long periods of torpor are interrupted by short periods of arousal with restoration of body temperature, heart and breathing rate, and blood pressure [6,7]. Indeed, hibernating mammals survive repeated cycles of I-R without injury. Regarding the kidneys, no significant functional or histopathologic changes were observed after the induction of torpor in the Syrian hamster (*Mesocricetus auratus*) [8]. Additionally, the kidneys of the dormouse (*Muscardinus avellanarius*) preserve their ultrastructure during hibernation and arousal [9].

Since warm ischemia-reoxygenation plays a pivotal role in the pathogenesis of many diseases, from a clinical point of view, the discovery of two different mouse-tailed bats species (*Rhinopoma microphyllum* and *Rhinopoma cystops*) that hibernate for five months in the high ambient temperature of geothermal caves is of great interest [10]. This demonstrates that shallow body temperature is not always necessary for resistance to I-R injury. Even more intriguing is the discovery of certain phylogenetically close to human primates such as the fat-tailed dwarf lemur (*Cheirogaleus medius*), which can hibernate and maintain warm body temperature at the same time [11,12]. In addition to the above observations, experimental studies have detected the resistance of various cell types and tissues from hibernators to warm I-R, remarking that the study of this phenomenon was exciting since warm I-R is involved in the pathogenesis of many human diseases [13,14]. Regarding the kidneys, warm anoxia-reoxygenation induces cell death to mouse (*Mus musculus*) renal tubular epithelial cells, while cells derived from the native hibernator Syrian hamster resist warm anoxia-reoxygenation-induced cell death [15]. Furthermore, hemorrhagic shock or cardiac arrest followed by resuscitation, situations that correspond to warm I-R, provoke significant functional renal impairment and pathological injury in rats (*Rattus norvegicus domesticus*), whereas arctic ground squirrels (*Urocitellus parryii*) are protected [16].

Studies in non-hibernating species have shown that death by lipid peroxidation, known also as ferroptosis, is the primary cause of renal tubular epithelial cell death during reperfusion or reoxygenation [17]. Initial experiments using various inhibitors identified ferroptosis as a new type of programmed cell necrosis. These experiments showed that inhibition of the cystine-glutamate antiporter (xCT) by impeding the entry of cystine into the cell, which is required for the synthesis of the antioxidant glutathione, led to ferroptosis [18]. Inhibition of ferroptosis by iron chelators indicated that intracellular labile bivalent iron is necessary for ferroptosis, most likely due to lipid membrane peroxidation through the Fenton reaction or iron-containing lipoxygenases [18]. Finally, ferroptosis also occurs during the inhibition of glutathione peroxidase 4 (GPX4), with the latter being the only GPX isoenzyme capable of reducing lipid hydroperoxides [18].

An elegant study with isolated renal tubules confirmed that during reoxygenation, cell death occurs through the ferroptotic pathway [17]. The same group has shown in vivo that another kind of cell death prevails, one that requires inflammatory cells, called necroptosis [19]. It is likely that ferroptosis of renal tubular epithelial cells is the primary event that leads to the release of damage-associated molecular patterns (DAMPs) and inflammation [20]. Subsequently, a second wave of necroptotic cell death follows. Thus, identifying ways to inhibit reoxygenation-induced ferroptosis may confer resistance to I-R injury. Indeed, a recent study showed that primary renal proximal tubular epithelial cells (RPTECs) of the native hibernator Syrian hamster resisted reoxygenation-induced cell death, whereas RPTECs derived from the phylogenetically close mouse died due to lipid peroxidation-induced cell death [15].

In this study, we evaluated the differences in the expression of specific ferroptotic pathway components in mouse and Syrian hamster RPTECs. More precisely, we assessed the effect of anoxia and subsequent reoxygenation on the expression of xCT, ferritin, which sequesters the toxic labile iron [18], and GPX4 in mouse and hamster RPTECs. Clarifying how hibernators escape ferroptosis may reveal clinically useful molecular targets for preventing or ameliorating I-R injury-induced human diseases or elongating the preservation of organ transplants.

## 2. Materials and Methods

### 2.1. Cells and Culture Conditions

Primary, passage one C57BL/6 mouse RPTECs (cat. no. C57-6015, Cell Biologics, Chicago, IL, USA) and primary, passage one Syrian hamster RPTECs (cat. no. HM-6015, Cell Biologics) were cultured in the Complete Epithelial Cell Medium kit, which is culture medium supplemented with epithelial cell growth supplement, antibiotics, and fetal bovine serum (cat. no. M6621, Cell Biologics). Cell Biologics warrants the standards of animal housing as well as the purity of the provided cells. Cells were expanded in 75 cm^2^ flasks and passage two cells were used for the experiments.

All cell cultures were performed at 37 °C. Cells were cultured in 6-well plates (300,000 cells/well) or 96-well plates (10,000 cells/well) for 12 h before the start of anoxia. Inverted microscopy confirmed that cell confluency did not differ at the onset of each experiment. To reduce the oxygen level to less than 1%, the GasPak^TM^ EZ Anaerobe Container System with Indicator (cat. no. 26001, BD Biosciences, S. Plainfield, NJ, USA) was used. These anoxic conditions imitated warm ischemia and lasted for 24 h.

After anoxia, cells were subjected to reoxygenation for 2 h. To imitate warm reperfusion, cells were washed with Dulbecco’s phosphate buffer saline (Sigma-Aldrich, Merck KGaA, Darmstadt, Germany), fresh supplemented culture medium was added, and the cells were cultured in a humidified atmosphere containing 5% CO_2_.

The above time points were selected according to a previous study that showed through cell imaging that primary Syrian hamster RPTECs were extremely resistant to both anoxia and reoxygenation, whereas primary mouse RPTECs deteriorated significantly after 48 h of anoxia or 4 h of reoxygenation [15]. In addition, in order to obtain reliable results, we chose to use primary cells instead of cell lines. For example, human embryonic kidney 293 (HEK293) cells, which are frequently used in I-R studies, are characterized, among others, by the absence of a functional p53 [21]. All experiments were repeated nine times.

### 2.2. Assessment of Cell Death and Lipid Peroxidation Due to Anoxia-Reoxygenation

To detect cell death, cells cultured in 96-well plates subjected to anoxia-reoxygenation as described above. After 2 h of reoxygenation, cell death was assessed by lactate dehydrogenase (LDH) release assay using the Cytotox non-radioactive cytotoxic assay kit (cat. no. G1780, Promega Corporation, Madison, WI, USA). Cell death was calculated by the equation Cell death (%) = (LDH in the supernatant/Total LDH) × 100. The experiments were performed in triplicate and repeated nine times.

To assess lipid peroxidation, cells were cultured in 6-well-plates, as described above. After 2 h of reoxygenation, the end product of lipid peroxidation, malondialdehyde (MDA), was measured fluorometrically in cell extracts with the lipid peroxidation (MDA) assay kit (cat. no. ab118970, Abcam, Cambridge, UK), which measures MDA indirectly. The MDA in the sample reacts with thiobarbituric acid (TBA) to generate a MDA-TBA adduct. The MDA-TBA adduct can be easily quantified colorimetrically (OD = 532 nm) or fluorometrically (Ex/Em = 532/553 nm). This assay detects MDA levels as low as 1 nmol/well colorimetrically and 0.1 nmol/well fluorometrically. A Bradford assay (Sigma-Aldrich, Merck KGaA) preceded the MDA measurement to adjust the lysate volume of each sample to an equal protein concentration.

### 2.3. Assessment of Cell Survival Due to Anoxia Reoxygenation and the Effect of α-Tocopherol

To evaluate whether α-tocopherol, a known factor that rescues cells from ferroptotic cell death [18,22], affects the survival of mouse RPTECs, cells cultured in 96-well plates were subjected to anoxia-reoxygenation as described above, but in the presence or not of 100 μM α-tocopherol (Sigma-Aldrich, Merck KGaA). Once the 2 h of reoxygenation passed, cell survival was assessed using the TACS XTT assay kit (Trevigen, Gaithersburg, MD, USA) according to the manufacturer’s protocol. Target cells were incubated with the 2,3-bis (2-methoxy-4-nitro-5-sulfophenyl)-5-[(phenylamino) carbonyl]-2H-tetrazolium hydroxide (XTT) reagent for 1 h. Cell survival was calculated by the equation Cell survival (%) = (XTT assay OD of the control/XTT assay OD of the evaluated condition) × 100. The experiments were performed in triplicate and repeated nine times.

### 2.4. Assessment of xCT, Ferritin and GPX4 Expression after Anoxia or Reoxygenation

Cells were cultured in 6-well plates as described previously. Once the 24-h anoxic period or the 2-h reoxygenation period was over, cells were lysed with the T-PER tissue protein extraction reagent (Thermo Fisher Scientific Inc., Waltham, MA, USA) supplemented with protease and phosphatase inhibitors (Sigma-Aldrich; Merck Millipore and Roche Diagnostics, Indianapolis, IN, USA, respectively). Protein was quantified via a Bradford assay (Sigma-Aldrich; Merck Millipore) and 10 μg from each sample was used for Western blotting.

Protein samples were electrophoresed at 180 V for 30 min in 4–12% bis-tris acrylamide gels (cat. no. NP0323BOX, NuPAGE 4–12% Bis-Tris Gel 1.0 mm × 15 well, Invitrogen; Thermo Fisher Scientific, Inc.). Blotting of the electrophoresed gel proteins on the polyvinylidene difluo-ride (PVDF) membrane was performed via electroporation at 30 V for 1 h. Skimmed milk in Tris-buffered saline with Tween-20 (Sigma-Aldrich; Merck Millipore) was used for blocking.

Blots were incubated with the primary antibody against the protein of interest for 16 h at 4 °C, followed by secondary antibody incubation for 30 min at room temperature. The Restore Western Blot Stripping Buffer (Thermo Fisher Scientific Inc.) was used whenever reprobing of the PVDF blots was performed.

Primary antibodies were the rabbit polyclonal antibody against the xCT (SLC7A11) (cat. no. ANT-111, Alomone Labs, Jerusalem, Israel), the mouse monoclonal antibody against the ferritin heavy (H) chain (cat. no. sc-376594, Santa Cruz Biotechnology, Santa Cruz, CA, USA), the mouse monoclonal antibody against the ferritin light (L) chain (cat. no. sc-74513, Santa Cruz Biotechnology), the mouse monoclonal antibody against GPX4 (cat. no. sc-166120, Santa Cruz Biotechnology), or the rabbit polyclonal antibody against β-actin (cat. no. 4967, Cell Signaling Technology, Cell Signaling Technology, Danvers, MA, USA). As secondary antibodies, an anti-rabbit IgG, HRP-linked antibody (cat. no. 7074, Cell Signaling Technology), or an anti-mouse IgG, HRP-linked antibody (cat. no. 7076, Cell Signaling Technology) were used.

Western blots bands were visualized by enhanced chemiluminescent detection using the LumiSensor Plus Chemiluminescent HRP substrate kit (GenScript Corporation, Piscataway, NJ, USA) and for densitometric analysis, the ImageJ software (National Institute of Health, Bethesda, MD, USA) was used.

### 2.5. Statistical Analysis

For statistical analysis, the IBM SPSS Statistics for Windows, Version 20 (IBM Corp., Armonk, NY, USA) was used. A Kolmogorov–Smirnov test confirmed the normal distribution of the evaluated variables. Independent samples t-test was used for the comparison of means between two variables and one-way ANOVA followed by the Bonferroni’s correction test for a comparison of means among more than two variables. Results were expressed as mean ± standard error of means (SEM) and a *p* < 0.05 was considered statistically significant.

To avoid the violation of the normal distribution of the variables when applying parametric statistical tests, in the analysis of the Western blot results, the *p* values were calculated by comparing the optical densities of the bands without normalization for the control group. However, in the manuscript, for the sake of simplicity, the results were presented and depicted after normalization of means for the control group.

## 3. Results

### 3.1. Anoxia-Reoxygenation Induces Lipid Peroxidation and Cell Death in Mouse RPTECs, but Not in Hamster RPTECs

In mouse RPTECs anoxia-reoxygenation induces cell death. Compared to the control cells, the LDH release assay revealed an increased cell death percentage by 285 ± 7% (*p* < 0.001) (Figure 1A). In contrast, hamster RPTECs resisted anoxia-reoxygenation, since the percentage of cell death was at 107 ± 4% of cell death detected in the control cells (*p* = 0.122) (Figure 1A).

Anoxia-reoxygenation increased lipid peroxidation in mouse RPTECs robustly. Compared to the control cells, 2 h of reoxygenation enhanced MDA levels by 2421 ± 105% (*p* < 0.001) (Figure 1B). In contrast, anoxia-reoxygenation did not induce lipid peroxidation in hamster RPTECs since cellular MDA was at 106 ± 3% of the level found in the control cells (*p* = 0.98) (Figure 1B).

### 3.2. α-Tocopherol Increases Cell Survival of Mouse RPTECs Subjected to Anoxia-Reoxygenation

In mouse RPTECs, anoxia-reoxygenation decreased cell survival at 45 ± 1% of the control (*p* < 0.001). The presence of α-tocopherol protected mouse RPTECs since the percentage of surviving cells subjected to anoxia-reoxygenation was 87 ± 1% of the control (*p* < 0.001 when compared to the control and *p* < 0.001 when compared to cells subjected to anoxia-reoxygenation in the absence of α-tocopherol) (Figure 2).

In hamster RPTECs, the presence of α-tocopherol did not affect cell survival. Cell survival of cells cultured without α-tocopherol after anoxia-reoxygenation was 99 ± 2% of the control (*p* = 1.0), whereas the presence of α-tocopherol cell survival was 100 ± 1% of the control (*p* = 1.0) (Figure 2).

### 3.3. Different Effect of Anoxia or Reoxygenation on the Expression of xCT, Ferritin, and GPX4 in Mouse and Syrian Hamster RPTECs

Compared to the control mouse RPTECs, anoxia increased xCT expression by a factor of 2.5 ± 0.3 (*p* = 0.001). However, reoxygenation did not alter xCT expression since it was 1.0 ± 0.1 of the level detected in the control cells (*p* = 1.0 when compared to the control cells and *p* < 0.001 when compared to the cells subjected only to anoxia) (Figure 3A and Figure 4A).

Anoxia increased xCT expression in hamster RPTECs 2.7 ± 0.5 times the expression found in the control cells (*p* < 0.001). Compared to the control hamster RPTECs, reoxygenation increased xCT expression further by a factor of 4.0 ± 0.7 (*p* < 0.001 when compared to the control cells and *p* < 0.001 when compared to the cells subjected only to anoxia) (Figure 3B and Figure 4A).

In mouse RPTECs, anoxia enhanced ferritin H level by a factor of 3.2 ± 0.3 (*p* < 0.001). Subsequent reoxygenation lowered the ferritin H level (*p* < 0.001), which remained 2.2 ± 0.2 times higher than in the control mouse RPTECs (*p* < 0.001) (Figure 3A and Figure 4B).

In hamster RPTECs, anoxia raised the ferritin H level by a factor of 2.6 ± 0.3 (*p* < 0.001). Subsequent reoxygenation augmented ferritin H expression further at 4.1 ± 0.4 times the level detected in control hamster RPTECs (*p* < 0.001 when compared to the control cells and *p* < 0.001 when compared to the cells subjected only to anoxia) (Figure 3B and Figure 4B).

Expression of ferritin L followed a similar trend in both mouse and hamster RPTECs. In mouse RPTECs, anoxia increased ferritin L by a factor of 3.5 ± 0.5 (*p* < 0.001). Compared to anoxia, reoxygenation decreased ferritin L (*p* < 0.001), which, however, compared to the control cells, remained higher by a factor of 2.5 ± 0.3 (*p* < 0.001) (Figure 3A and Figure 4C).

In hamster RPTECs, ferritin L level after anoxia was 4.6 ± 0.6 times the level detected in the control cells (*p* < 0.001). Reoxygenation increased the ferritin L level further by 6.1 ± 0.7 times the level found in control hamster RPTECs (*p* < 0.001 when compared to the control cells and *p* < 0.001 when compared to the cells subjected only to anoxia) (Figure 3B and Figure 4C).

In mouse RPTECs, anoxia raised the GPX4 level by a factor of 1.8 ± 0.2 (*p* = 0.014). Reoxygenation increased the GPX4 level further by 4.0 ± 0.7 times the level detected in the control mouse RPTECs (*p* < 0.001 when compared to the control cells and *p* < 0.001 when compared to the cells subjected only to anoxia) (Figure 3A and Figure 4D).

In hamster RPTECs, anoxia also enhanced GPX4 expression by a factor of 3.6 ± 0.3 (*p* < 0.001). However, subsequent reoxygenation lowered the GPX4 level by 2.0 ± 0.2 times the level observed in the control hamster RPTECs (*p* < 0.001 when compared to the control cells and *p* < 0.001 when compared to the cells subjected only to anoxia) (Figure 4B,D).

## 4. Discussion

Clarifying how hibernators escape I-R injury may confer new molecular targets for preventing or ameliorating various I-R-induced human diseases as well as for elongating the preservation of organ transplants.

Recapitulating the results of a previous study [15], this study showed that after warm anoxia-reoxygenation and contrary to mouse RPTECs, hamster RPTECs did not suffer lipid peroxidation and resisted cell death. In addition, we showed that α-tocopherol, a known factor that rescues cells from ferroptotic cell death [18,22], enhanced the survival of mouse RPTECs subjected to anoxia-reoxygenation. These results indicate that the massive ROS production after reoxygenation induces ferroptosis or cell death by lipid peroxidation in the context of mouse RPTECs, whereas hamster RPTECs exhibited no susceptibility to such deleterious effects. In addition to this, and considering that under the same experimental conditions, reoxygenation does not provoke other main types of cell death such as apoptosis, necroptosis, or autophagic cell death in mouse RPTECs [15], it is reasonable to assume that hamster RPTECs tolerate reoxygenation due to adaptations of the ferroptotic pathway. As already noted, experiments in freshly isolated mouse renal tubules revealed that ferroptosis is the primary cause of cell death due to warm anoxia-reoxygenation in this non-hibernator [17].

Following this, we evaluated the differences in the expression of specific significant components of the ferroptotic pathway [18] in mouse and hamster RPTECs subjected to anoxia or subsequent reoxygenation, which are depicted in Figure 5.

Inhibition of GPX4 induces ferroptosis. CPX4 is the only known enzyme capable of reducing cell membrane polyunsaturated phospholipid hydroperoxides [18]. Mice with GPX4 under-expression are incredibly vulnerable to ferroptosis-mediated AKI [23]. To our knowledge, the expression or the activity of this specific GPX isoenzyme has not been evaluated in hibernating species yet. In our experimental model, anoxia increased GPX4 expression in mouse and to a greater extent in hamster RPTECs, preparing the cells for the subsequent reoxygenation-induced ROS overproduction. However, and contrary to our expectations, reoxygenation further enhanced GPX4 expression in mouse RPTECs, whereas it decreased GPX4, albeit to a level higher than the control, in hamster RPTECs. Thus, alterations in GPX4 level are not responsible for the resistance of hamster RPTECs from anoxia-reoxygenation-induced ferroptosis.

Interestingly, these GPX4 alterations in both mouse and hamster RPTECs followed the same fluctuations as observed in similar experimental conditions regarding the level of the endoplasmic reticulum (ER) stress sensor phosphorylated protein kinase RNA-like ER kinase (pPERK) [24], and of the phosphorylated eukaryotic translation initiation factor 2α (peIF2α) [25]. Thus, it is possible that the observed GPX4 alterations in mouse and hamster RPTECs under anoxia or reoxygenation of the present study may result from corresponding changes of the pPERK-peIF2α-activating transcription factor 4 (ATF4)-heatshock 70-kDa protein 5 (HSPA5) axis since HSPA5 binds to GPX4 and protects it from degradation [26].

Treatment of cells with iron chelators prevents ferroptosis indicating that labile intracellular iron is required for membrane polyunsaturated phospholipid peroxidation via the Fenton reaction or iron-containing lipoxygenases [18]. Ferritin can sequester the toxic bivalent labile iron in its non-toxic trivalent form through the ferroxidase activity of the ferritin H subunit [18]. Overexpression of ferritin H in mouse kidneys confers protection against I-R-induced AKI [27]. In our experimental model, anoxia increased both ferritin H and ferritin L subunits in both mouse and hamster RPTECs. However, during reoxygenation, the ferritin subunits expression levels further increased in hamster RPTECs, whereas in mouse RPTECs, the levels were found decreased, albeit to a level higher than the control. Hence, it seems that ferritin may be a ferroptotic pathway factor that confers protection to hamster RPTECs from warm anoxia-reoxygenation. Interestingly, the primate hibernator *Cheirogaleus crossleyi* may be protected from iron-catalyzed oxidative damage by upregulating ferritin genes to store the labile bivalent iron in the non-toxic trivalent form [28].

Notably, anoxia augments ferritin expression [29], while oxidative stress enhances its degradation, releasing the toxic iron and further increasing the oxidative stress through the Fenton reaction [30]. As a consequence, it is possible that the decrease in ferritin levels in the mouse RPTECs during reoxygenation may indicate increased oxidative stress in these cells when compared to the hamster RPTECs. The high blood antioxidants such as glutathione concentration and catalase, total GPX, and superoxide dismutase activities observed during interbout arousal in Syrian hamsters [31,32], which correspond to the reoxygenation phase of our experiments, support this possibility.

Regarding ferroptosis, among the various antioxidant systems that may confer hamster RPTECs resistance to oxidative stress, glutathione is of particular interest. Inhibition of the cystine-glutamate antiporter xCT induces ferroptosis by depleting intracellular glutathione, which is required for the reduction of cell membrane phospholipid hydroperoxides by GPX4 [18]. It should be noted that glutathione is also necessary for the reduction of hydrogen peroxide by the abundant cytosolic GPX1 [33]. Thus, besides its role in the reduction of cell membrane phospholipid hydroperoxides, xCT affects the total oxidative status of the cell. Regarding the kidneys, xCT-deficient mice are more sensitive to I-R-induced AKI [34]. However, to our knowledge, xCT expression has not yet been evaluated in hibernating mammals. In the present experimental model, anoxia increased xCT expression in both mouse and hamster RPTECs. Under reoxygenation, at which ROS overproduction occurs, xCT expression was further increased in hamster RPTECs, but decreased almost to the control levels in mouse RPTECs. Hence, it is very likely that the increased expression of the ferroptotic pathway element xCT in hamster RPTECs may contribute to the resistance of these cells to warm ischemia-reoxygenation.

Interestingly, the peIF2α-ATF4 axis upregulates xCT [35], and in an experimental model similar to the present study, increased peIF2α levels were observed under anoxia in both mouse and hamster RPTECs [25]. However, in that study, in hamster RPTECs, reoxygenation significantly decreased peIF2α below the control levels, while xCT expression was significantly elevated. Another attractive mechanism for xCT upregulation during reoxygenation in hamster RPTECs could rely on is the transcription factor nuclear factor erythroid 2 (NF-E2)-related factor 2 (Nrf2) since this factor upregulates xCT [36,37] as well as many other antioxidants [38]. Increased Nrf2 activity has been detected in many organs of hibernating rodents during interbout arousals, though in the context of kidneys, the activity of this transcription factor was significantly decreased [39,40]. Clarifying the role of Nrf2 in the resistance of hibernators to warm anoxia-reoxygenation is a key concept, and we are currently evaluating this in vitro and in vivo. In the mouse brain subjected to I-R, the transcription factor hypoxia-inducible factor 1α (HIF-1α) is upregulated and increases xCT expression [41]. However, in our experimental model, xCT decreased during reoxygenation in mouse RPTECs, and HIF-1α kinetics in the kidneys of hibernating mammals during torpor or interbout arousals have also not been evaluated yet. Thus, the exact molecular mechanisms responsible for alterations in xCT expression between mouse and hamster RPTECs remain to be elucidated.

A limitation of our study is that although we assessed the key initially discovered elements of the ferroptotic pathway [18], we did not investigate all of the factors that may affect ferroptosis. For instance, changes of the cell membrane structure toward a higher percentage of polyunsaturated phospholipids may increase the sensitivity to lipid peroxidation and ferroptosis [18], and cell membrane structure alters during hibernation [42]. Related to the above context, another probable therapeutic target is the enzyme acyl-CoA synthetase long-chain family member 4 (ACSL4), which sensitizes cells to ferroptosis by enriching cell membranes with long polyunsaturated ω3 fatty acids [43]. Certainly, there are factors such as the trassulfuration pathway that synthesizes cysteine, glutaminolysis, lipoxygenases, and many others [18], which remain to be evaluated.

Finally, the in vitro nature of our study is another limitation since direct predictions from an in vitro study to an in vivo model are not always safe. Thus, our study could be considered as a starting point for further investigation of the mechanisms that confer resistance to I-R-induced ferroptosis in hibernators. However, sometimes the simplicity of in vitro and ex vivo experiments may reveal the right sequence of events that take place in vivo. The documented sequence of ischemia-reperfusion-induced ferroptosis, ferroptosis-induced inflammation, and inflammation-induced necroptosis in mouse renal tubules is an excellent example [17,19].

## 5. Conclusions

In conclusion, hamster RPTECs resist warm anoxia-reoxygenation-induced ferroptosis. Expression levels of GPX4, ferritin, and xCT, components of the ferroptotic pathway, were evaluated, with the last two being more likely to confer tolerance to I-R. Thus, interfering with the expression of ferritin or the expression and activity of the bidirectional xCT antiporter may prevent or ameliorate various I-R-induced human diseases including AKI as well as the deterioration of organ transplants during their preservation. The study of hibernating species that have been enriched throughout evolution with adaptable strategies against I-R injury may eventually provide solutions for human I-R injury-induced diseases.

## Figures and Tables

**Figure 1 biology-08-00022-f001:**
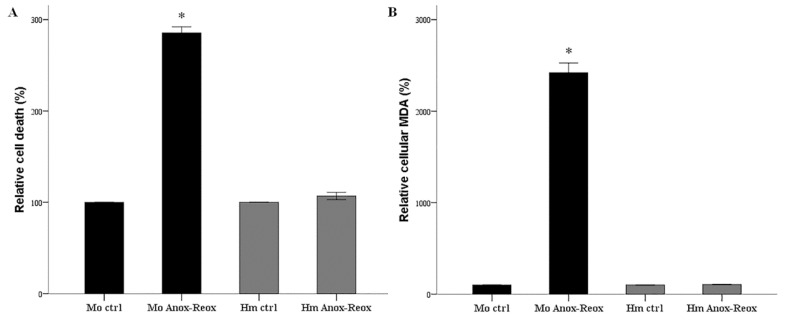
Cell death and lipid peroxidation due to warm anoxia-reoxygenation. LDH release assay revealed that warm anoxia-reoxygenation induced cell death in mouse RPTECs, whereas hamster RPTECs tolerated 24 h of anoxia followed by 2 h of reoxygenation. These experiments were repeated nine times in triplicate (**A**). Measurement of cellular MDA revealed that in mouse RPTECs, warm anoxia-reoxygenation caused robust lipid peroxidation. On the contrary, no lipid peroxidation was detected in hamster RPTECs. These experiments were performed nine times (**B**). Error bars correspond to SEM and the asterisk (*) to a *p* < 0.001 when compared to the respective control cells. Mo stands for mouse, Hm for hamster, ctrl for control, and Anox-Reox for anoxia-reoxygenation.

**Figure 2 biology-08-00022-f002:**
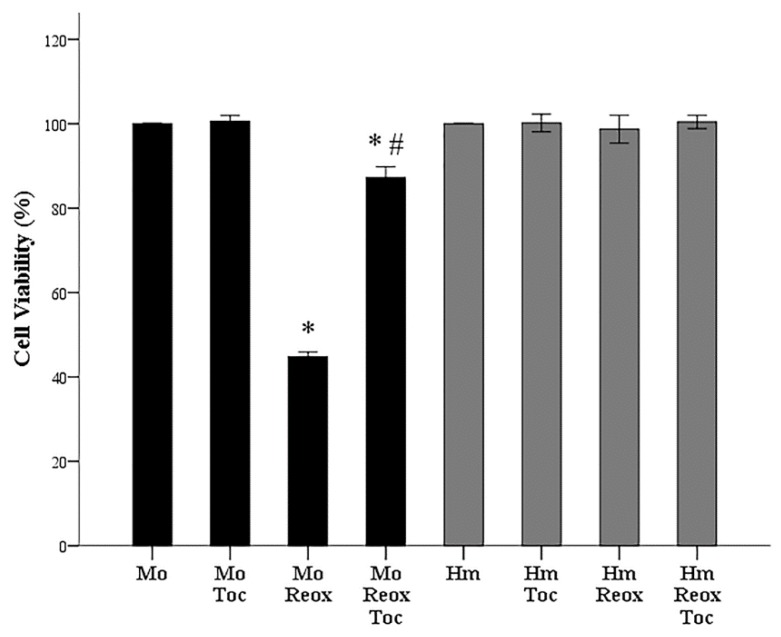
Cell survival due to warm anoxia-reoxygenation and the effect of α-tocopherol. The XTT assay revealed that cell survival in mouse RPTECs was decreased in contrast to hamster RPTECs that well tolerated 24 h of anoxia followed by 2 h of reoxygenation. The presence of α-tocopherol (Toc) did not affect the cell viability of hamster RPTECs. However, it significantly enhanced cell viability of mouse RPTECs subjected to anoxia-reoxygenation. These experiments were repeated nine times in triplicate. Error bars correspond to SEM, the asterisk (*) to *p* < 0.001 when compared to the respective control cells, and the hashtag (#) to *p* < 0.001 between cells subjected to anoxia-reoxygenation without or with α-tocopherol. Mo stands for mouse, Hm for hamster, ctrl for control, Reox for reoxygenation, and Toc for α-tocopherol.

**Figure 3 biology-08-00022-f003:**
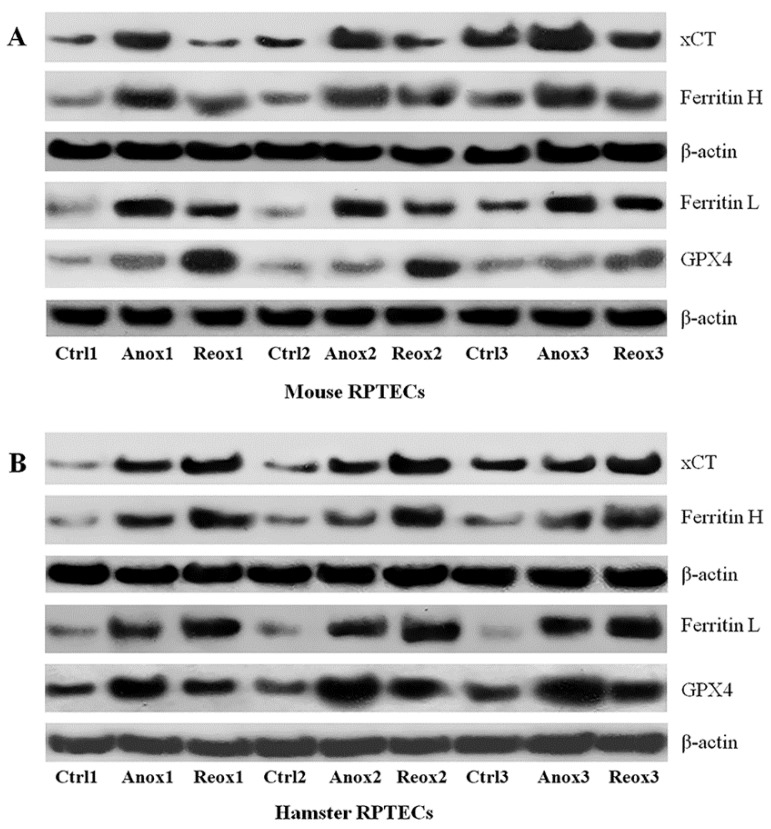
Western blots from mouse and hamster RPTECs subjected to anoxia or reoxygenation. RPTECs were subjected to 24 h of anoxia or 24 h of anoxia followed by reoxygenation for 2 h. These experiments were performed nine times. Panel (**A**) represents the Western blots for xCT, ferritin H, ferritin L, and GPX4 from three representative experiments in mouse RPTECs. Panel (**B**) represents the same blots in hamster RPTECs. Ctrl stands for control, Anox for anoxia, and Reox for reoxygenation.

**Figure 4 biology-08-00022-f004:**
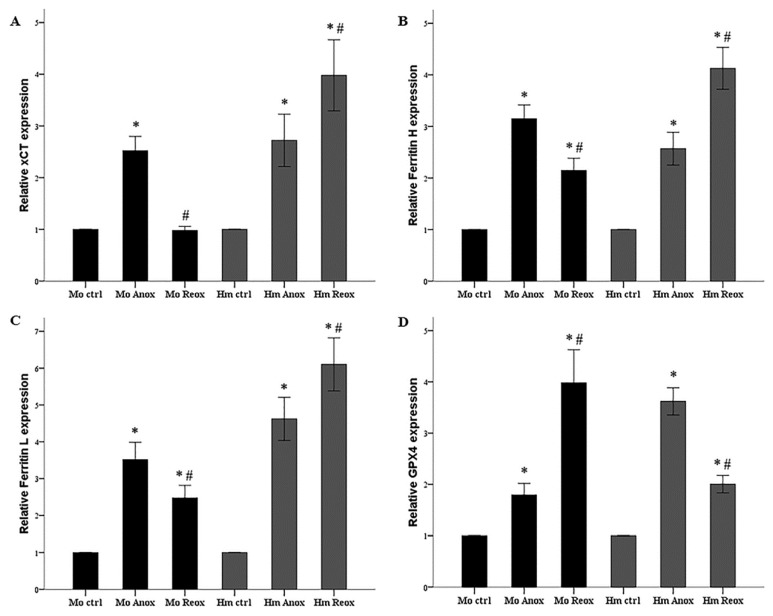
Expression of xCT, ferritin H, ferritin L and GPX4 in mouse and hamster RPTECs subjected to anoxia or reoxygenation. In mouse RPTECs, anoxia enhanced the levels of xCT, ferritin H, ferritin L, and GPX4 (panels **A**, **B**, **C**, and **D** respectively). During reoxygenation, at the time of the increased reactive oxygen species production, xCT, ferritin H, and ferritin L decreased, whereas GPX4 increased (panels **A**, **B**, **C**, and **D** respectively). In hamster RPTECs, anoxia also increased the levels of xCT, ferritin H, ferritin L, and GPX4 (panels **A**, **B**, **C**, and **D** respectively). However, reoxygenation raised xCT, ferritin H, and ferritin L further, but lowered GPX4 (panels **A**, **B**, **C**, and **D** respectively). These experiments were repeated nine times. Error bars correspond to SEM. Except for the asterisk (*) that detected the significance of GPX4 increase in mouse RPTECs in panel (**D**) and corresponds to a *p* < 0.05, in all other cases, the asterisk (*) corresponds to *p* < 0.001 when compared to the respective control cells and the hashtag (#) to *p* < 0.001 between cells subjected only to anoxia and cells subjected to anoxia followed by reoxygenation. Mo stands for mouse, Hm for hamster, ctrl for control, Anox for anoxia, and Reox for reoxygenation.

**Figure 5 biology-08-00022-f005:**
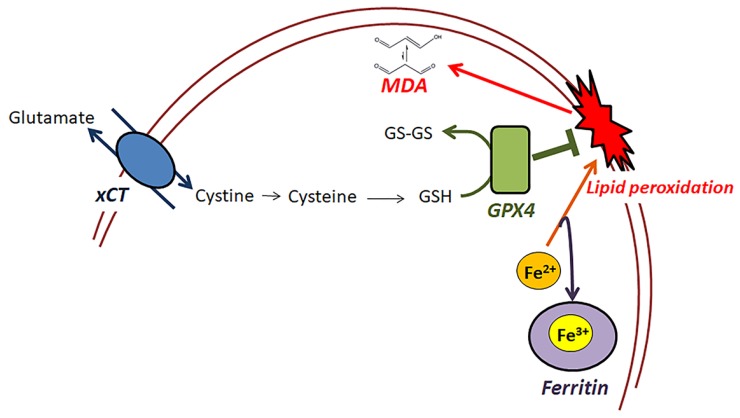
Components of the ferroptotic pathway evaluated in the present study. In this study, the effect of warm anoxia-reoxygenation on the main initially discovered elements of the ferroptotic pathway was evaluated in RPTECs derived from mouse or the native hibernator Syrian hamster. The cystine-glutamine antiporter (xCT) facilitates the transport of cystine into the cells, which is required for the synthesis of glutathione (GSH). Glutathione peroxidase 4 (GPX4) is the only GPX isoenzyme capable of reducing cell membrane polyunsaturated phospholipid hydroperoxides, and in this reaction, it oxidizes GSH. Labile bivalent iron (Fe^2+^) is necessary for cell membrane lipid peroxidation, while ferritin stores the labile bivalent iron in its nontoxic trivalent form (Fe^3+^) preventing lipid peroxidation. Our study showed that the lack of lipid peroxidation, assessed by the cellular malondialdehyde (MDA), in hamster RPTECs subjected to warm anoxia-reoxygenation might result from the increased xCT and ferritin expression.

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
