# Peer review of "Factors that May Protect the Native Hibernator Syrian Hamster Renal Tubular Epithelial Cells from Ferroptosis Due to Warm Anoxia-Reoxygenation"

_biology, 2019, doi:10.3390/biology8020022_

Round 1
Reviewer 1 Report
The authors present here an in vitro study comparing the protein expression levels of 3 genes involved in ferroptosis (or its inhibition) in murine versus hibernating hamster proximal epithelial cells. They demonstrate that hamster cells are likely to be protected from anoxia induced ferroptosis via ferritin or xCT upregulation. The data is sound and consistent, however one concern needs to be addressed.
1) Although the authors are the first to demonstrate expression levels of several (3) components of the ferroptotic pathway in hibernators, the data load of the manuscript, in my opinion, is too limited to warrant publication. The authors do argument the limitations as well as the relevance of conducting in vitro research, however, their model could be exploited more. In vitro systems offer the opportunity, as the authors acknowledge, to dissect molecular mechanisms in a more profound way. I suggest efforts should be taken in that direction to generate additional data (gene knock-down, use of inhibitors of the cell’s protection against ferroptosis and thereby sensitize the hamster cells, additional gene expression analysis, focus on the behavior and upstream signals of transcription factors Nrf2, peIF2alpha, …).
2) An in vivo experiment of tremendous clinical relevance would be to perform a renal ischemia/reperfusion experiment in hibernating hamsters to verify whether kidneys are protected from an insult that is modelling renal transplantation. Given the introduction, it appears that such an experiment has not yet been performed.
Author Response
Response to the Reviewer 1
Firstly, we would like to express our appreciation for the good comments of the Reviewer about our study, as well as for his/her suggestions for future research in the field.
1. According to the comments, we added an additional experiment in the revised manuscript in which cell survival was assessed in mouse and hamster RPTECs subjected to reoxygenation in the presence or not of the ferroptosis inhibitor vitamin E. Vitamin E increased cell survival of mouse RPTECs confirming the ferroptotic nature of their death. Abstract, Methods, Results, Discussion, and Figures are revised accordingly.
Regarding the peIF2α, we have already assessed its level under the same experimental conditions in a previous study (Eleftheriadis et al., Biomed Rep. 2018; 9:503-510), which is already cited in the manuscript. As noted, in that study xCT alterations did not follow peIF2α changes in hamster cells subjected to reoxygenation.
Regarding the Nrf2. Although, as already noted in the manuscript, Nrf2 activity decreases in the kidney of hibernators during interbout arousals, the suggested notion of Nrf2 pathway involvement is extremely interesting. We are currently evaluating the oxidative stress//cystathionine-β-synthase/cystathionine-γ-lyase//Nrf2 pathway both in vitro and in vivo.
2. We strongly agree with the suggested experiment, but since this is a new field for investigation our focus in this study was to aprehend the molecular mechanisms that protect hibernators from warm reoxygenation before proceeding to expensive and animal sacrificing in vivo studies. We are currently evaluating the effect of oxidative stress // cystathionine-β-synthase/cystathionine-γ-lyase // Nrf2 pathway on ferroptosis both in vitro and in vivo. A note has been added in the revised manuscript.
3. In the revised manuscript a thorough spell-check was performed.
Reviewer 2 Report
In the work of Eleftheriadis et al., the authors study potential factor regulating increase sensitivity to ferroptosis in mouse proximal renal tubular epithelial cells compared to the native hibernator Syrian hamster (SH). The authors also propose that the increased resistance of the SH cells is due to up-regulation of antiferroptotic genes. The study is overall interesting and present a differential response between 2 different species to the process of hypoxia/re-oxygenation. Yet one would need to be careful when drawing strong conclusion from this comparison due to technical points on how the animals were house, how cells were isolated etc. Nonetheless excluding these points I would like to list other short comings that deserve particular attention in order to strengthen the current work.
There instance were the data is not satisfactorily present and deserve better clarifications.
Figure 1A – do not use relative cell (%), if out 10.000 cells you have 1 dying in the control and 3 in the reoxygenation would this be meaningfull? It sure looks good on a graph but the meaning is not clear. Please us % of surviving cells.
Figura 1B – the authors claim to measure MDA – the kit used is actually measuring thiobarbituric acid reactive species (TBARS) this is not MDA, to claim this the authors should have separated by HPLC and measure the MDA-TBA peak.
Would be important to compare on the same blot mouse and hamster cells in order to have an idea of their levels. Also it would be helpful to address the comparative response to nrf2 activators response in both cell lines in order to draw better conclusion.
Moreover, there is no compelling case that ferroptosis is taking place – the authors should provide evidence that in the re-oxygenation model, ferroptosis inhibitors are blocking cell death and lipid peroxidation. Similarly, if lipid peroxidation has a causative role vitamins such as aToc should be able to provide rescue.
Author Response
Response to the Reviewer 2
Initially, we would like to thank the Reviewer since his/her valuable comments helped us to improve our study and manuscript.
1. Regarding the primary mouse and hamster RPTECs, we did not isolate those cells. We used commercially available cells from a specialized company (Cell Biologics, Chicago, IL, USA) which warrants the standards of animal housing, as well as the purity of the cells. This is noted in the revised manuscript.
2. The sensitivity of the LDH release assay is not that narrow. However, we agree with the Reviewer’s comment that cell survival provides a better image of the phenomenon. Thus, in the revised manuscript cell viability was also assessed via the XTT assay. Abstract, Methods, Results, Discussion, and Figures are revised accordingly.
3. The kit measures MDA indirectly. The MDA in the sample reacts with thiobarbituric acid (TBA) to generate an MDA-TBA adduct. The MDA-TBA adduct can be easily quantified colorimetrically (OD = 532 nm) or fluorometrically (Ex/Em = 532/553 nm). This assay detects MDA levels as low as 1 nmol/well colorimetrically and 0.1 nmol/well fluorometrically. This indirect measurement and the principle of the assay are described in more detail in the revised manuscript.
4. Comparing the OD values between electrophoresed protein bands from different species in the same western blotting gel is not accurate. Small differences in their amino acid sequence (or glycosylation) may affect significantly the affinity of the primary antibody used in western blotting which makes such a comparison impossible.
5. We have already clarified and published some molecular pathways that take place in this new field of hibernators’ resistance to warm anoxia-reoxygenation. Although, as already noted in the manuscript, Nrf2 activity decreases in the kidney of hibernators during interbout arousals, the suggested notion of Nrf2 pathway involvement is extremely interesting. Currently, we are working on the effect of oxidative stress // cystathionine-β-synthase/cystathionine-γ-lyase // Nrf2 pathway on ferroptosis both in vitro and in vivo. A note has been added in the revised manuscript.
6. This comment is critical. Hence, according to the Reviewer’s suggestion, we added an additional experiment in the revised manuscript in which cell survival was assessed in mouse and hamster RPTECs subjected to reoxygenation in the presence or not of vitamin E, which rescues cells from ferroptosis. Vitamin E increased cell survival of mouse RPTECs confirming the ferroptotic nature of their death. Abstract, Methods, Results, Discussion, and Figures are revised accordingly.
7. English editing applied in the revised manuscript.
Reviewer 3 Report
The authors nicely summarize state-of-the-art information regarding (mainly) regulation of the ferroptotic pathway. They describe and investigate a very intersting phenomenon: the difference of hibernating and non-hibernating species with regard to different regulation of the ferroptosis pathway (which results in an escape strategy of hibernating animals from I/R induced tissue injury). Analysing anoxia and anoxia/reperfusion treated primary mouse and hamster RPTECs the authors find different regulation of xCT, ferritin and GPX4 regarding the species. The experiments are well designed and presented clearly. However, the interpretation of the data is in my point of view a bit to exaggerated. All components of the ferroptotic pathway are known to be promising clinically relevant targets anyway, In addition the authors do not mention other known and well investigated regulators of ferroptosis, e.g. ACSL4.
Minor revisions: The manuscript needs some grammatical correction (e.g. page 1, line 15, my --> by)
Author Response
Response to the Reviewer 3
First of all, we would like to thank the Reviewer for his/her encouraging comments.
1. We evaluated some key (and firstly discovered) components of the ferroptotic pathway. The limitation of not assessing all the elements that directly or indirectly affect ferroptosis, such as the role of ACSL4 which sensitizes cells to ferroptosis by enriching cell membranes with long polyunsaturated ω3 fatty acids or the role of transsulfuration pathway in preventing ferroptosis, has been added in the limitations of the revised manuscript with the related citations.
2. The revised manuscript was re-checked for language errors.
Round 2
Reviewer 1 Report
The manuscript improved from the review process. The addition of the Vitamin E experiment and limitations is appreciated. Yet, I need to be honest and still feel that the overall amount of data is limited. Nonetheless, if the editors feel that this well performed study (although concise) is acceptable within the current context of "Biology" while it is being tracked for impact factor, I will comply to their decision.
Author Response
We would like to thank the reviewer since his/her valuable comments helped us to improve our manuscript.
Reviewer 2 Report
nothing to state
Author Response
We would like to thank the reviewer since his/her comments helped us to improve our manuscript.